# Persistent Left Superior Vena Cava Significance in Prenatal Diagnosis—Case Series

**DOI:** 10.3390/jcm11144020

**Published:** 2022-07-12

**Authors:** Mircea-Octavian Poenaru, Bashar Haj Hamoud, Romina-Marina Sima, Ionut-Didel Valcea, Radu Chicea, Liana Ples

**Affiliations:** 1Department of Obstetrics and Gynecology, The ‘Carol Davila’ University of Medicine and Pharmacy, 020021 Bucharest, Romania; mircea.poenaru@umfcd.ro (M.-O.P.); liana.ples@umfcd.ro (L.P.); 2The “Bucur” Maternity—‘Saint John’ Hospital, 040294 Bucharest, Romania; ionuvalcea@yahoo.com; 3Department for Gynecology, Obstetrics and Reproductive Medicine, Saarland University Hospital, 66421 Homburg, Germany; bashar.hajhamoud@uks.eu; 4Faculty of Medicine, ‘Lucian Blaga’ University of Sibiu, 550024 Sibiu, Romania; radu.chicea@gmail.com

**Keywords:** congenital heart disease, persistent left superior vena cava, prenatal diagnosis, congenital venous anomaly

## Abstract

The persistent left superior vena cava (PLSVC) is a congenital heart anomaly reported in 0.3–0.5% of the general population and can be associated with congenital heart diseases in up to 8% of cases. Prenatal identification of PLSVC is important to prompt an extended cardiac and extracardiac fetal examination. We retrospectively reevaluated anomaly scans performed in our unit in a 2-year interval according to the national guidelines to evaluate the incidence of PLSVC and its association with prenatal morbidity. In our population, the incidence of PLSVC was 0.31%, and we found a low association with cardiac and extracardiac anomalies. The standard sections (three-vessel and trachea view, four-chamber view and outflow tract’s view) are insufficient to exclude cardiac anomalies whenever PLSVC is found. In our case series, only one newborn required postnatal surgery for total pulmonary vein anomaly, and at 2 years of life all babies had a normal evolution. Prenatal diagnosis of PLSVC can raise counseling issues; therefore, awareness of its good outcome when isolated and need for an extended examination to rule out other anomalies is very important.

## 1. Introduction

Persistent left superior vena cava (PLSVC) is a frequent congenital heart anomaly reported in 0.3–0.5% of the general population. Generally, is a benign finding, without hemodynamic consequences, but its association with congenital heart diseases (CHD) in 4–8% is significant and requires a detailed prenatal cardiac evaluation and a systematic follow-up of the apparently isolated PLSVC to exclude unfavorable outcomes [1,2].

Cardio-vascular system development starts early in embryonic life, and around the 23rd day the primitive heart starts beating. The embryonic venous system is formed from three pairs of primitive veins, i.e., the vitelline, umbilical and cardinal veins. There are four cardinal veins, i.e., two pairs of anterior and posterior (left and right) veins that are responsible for the systemic venous return. In week 7, the innominate vein (left brachycephalic vein) is formed by the anastomosis of the left and right anterior cardinal veins. From the anterior part, the internal jugular veins are derived, and from the inferior part, the superior vena cava is formed. The pair of posterior cardinal veins are responsible for inferior vena cava and iliac veins appearance (Figure 1). In some cases, the right superior vena cava is missing, and the venous blood from the upper part of the body and head returns to the right atrium through the PLSVC, which drains in the coronary sinus [3].

The isolated PLSVC has no hemodynamic consequence, even when the superior right vena cava is missing, since the venous blood from the upper part is drained in the right atrium. Considering the increased amount of blood accommodated through the coronary sinus (CS), the PLSVC is always, in variable degrees, associated with CS dilatation [4].

There are is mention of social or environmental factors that could be responsible for PLSVC in the literature, which otherwise is scarce regarding that condition, reporting studies mainly regarding its prenatal diagnosis. Postnatal studies focus on the importance of being aware of that anomaly at the time of cardiac surgery (i.e., mitral surgery) or when it is associated with other cardiac anomalies such as heterotaxy, aortic coarctation absence of Right Superior Vena cava and dilated coronary sinus. Awareness of PLSVC existence is important whenever an interventional procedure requires venous cardiac catheterization (i.e., pacemaker implantation), or when associated cardiac or extracardiac anomalies are present [4,5,6].

The aim of this study was to evaluate the incidence and significance of the prenatal diagnosis of PLSVC in our clinic and to review the literature data regarding PLSVC diagnosis and association with cardiac and extracardiac anomalies.

## 2. Materials and Methods

We conducted a retrospective study on the ultrasound scan anomalies performed in our unit in the last 2 years. The study started in February 2017 and ended in March 2019 due to the COVID 19 pandemic, when our section was closed for general obstetrics. As a tertiary maternity center, we receive referred pregnancies from the southern part of Romania (4 administrative departments and usually from Bucharest, which means that our main group of patients are from urban areas. In our country, the population is very homogenous, consisting mainly of Caucasian people. No other races were encountered in our study). The ultrasound scans were performed according to the National Protocols approved by the Ministry of Health during the second (20–23 weeks) and the third trimester (and 30–32 weeks) by two fetal medicine certified specialists, using equipment with a probe of 3.5 MHz. The patients were informed about the scans prior to examination and provided consent for examination and medical data use for research and teaching purpose. Standard sections of the heart obtained from all patients included: thoracal transverse sections in 4-chamber view, three-vessel and trachea view (TTVV) and images of the left and right outflow tract. As a teaching and referral center, we also usually perform a thorough cardiac assessment including a sagittal view of the arches and a bicaval section. The PLSVC is diagnosed in three main sections: 3-vessel view, where a small additional vessel is found at the main pulmonary artery (MPA) left, 4-chamber view (4 CV), where the PLSVC is detected as a round shape adjacent to the wall of the left atrium and a dilated coronary sinus is present, and in a sagittal view that shows the “pipe”-like aspect of the PLSVC. Whenever a suspect image is found, an extracardiac detailed scan is also performed. Images and clips are stored for further examination and as a proof if required. The cardiac images were reviewed, and for the cases with PLSVC, we retrieved from the patients file the following data: mothers’ age, pregnancy age, parity, history of previous congenital anomalies, cardiac or extracardiac associated anomalies, pregnancy and fetal outcome, age and type of delivery, neonate outcome and baby evolution at 12 months of life.

## 3. Results

In 24 months, 1897 pregnancies were scanned according to our records, and 6 cases of PLSVC were identified (incidence of 0.31%) (Table 1). Two cases were referred for a second opinion (case 1 and 4). We encountered only one obese (BMI 31.4 kg/m^2^) patient, while three patients were overweight (BMI over 25 kg/m^2^). No correlation between smoking status and incidence of PLSVC could be made. The main characteristics of the cases are indicated in Table 1.

## 4. Cases—Ultrasound Findings Description

### 4.1. Case 1

This case, a 23 years primigravida scanned at 22 + 1 weeks, presented the most commonly encountered and typical appearance of PLSVC. The diagnosis was made on four main images, three-vessel and trachea view (Figure 2), where four vessels instead of three could be see: 4CV left atrium wall with dilated coronary sinus (Figure 3), sagittal image depicting the “pipe sign” (Figure 4) and the absence of the innominate vein (Figure 5). Since there was no cardiac anomaly, but minor additional markers were found (renal pelvis dilatation and single umbilical artery), a karyotype was recommended and performed, with normal results. The findings were confirmed at birth by a pediatric cardiologic examination and interpreted as normal with good outcome.

### 4.2. Case 2

This case was an uncommon situation defined by the association of PLSVC with the absence of RSVC. A 33-year-ols secundigravida was scanned at 21 weeks of pregnancy. In the three-vessel trachea view, PLSVC was identified, but no RSVC was observed (Figure 6); a dilated coronary sinus was also identified (Figure 7) and, typically for this situation, the innominate vein showed an inverted flow (Figure 8). On a sagittal section, the diagnosis was confirmed, as the “pipe sign” was identified (Figure 9). At 32 weeks and postnatally, the diagnosis was reconfirmed, and the baby had an uneventful evolution until 1 year of life.

### 4.3. Case 3

Case 3 was found in a dichorionic diamniotic pregnancy of a 29-year-old primipara. One fetus had multiple venous anomalies: PLSVC, absent infradiaphragmatic IVC and hemiazygos vein (Figure 10), absent RSVC (Figure 11), drainage of the hemiazygos in PSLVC (Figure 12) and dilated CS (Figure 13). The pregnancy was uneventful until a premature birth occurred at 35 weeks; the baby is doing well at 2 years.

### 4.4. Case 4

This case underlines the importance of careful examination and follow-up, whenever an isolated PSLVC is found. In this case, a 34-year-old woman was scanned at 22 + 1 weeks in another center and referred at 26 weeks to our unit for isolated PSLVC (Figure 14). We identified, beside PSLVC, the innominate vein with a normal flow (Figure 15) and the collector vessel behind the left atrium directed to the IV (Figure 16 and Figure 17). Extended cardiac examination was performed in order to rule out pulmonary vein anomaly, and we considered that at least two pulmonary veins were connected to the left atrium. At 32 weeks, the patient was rescanned, and the images suggested a partial pulmonary vein return anomaly and “cor triatriatum”. Postpartum, the baby was transferred to a cardiac surgery unit where he underwent surgery at 14 days of life. At 9 months, the baby is doing well.

No particular image was available for cases 5 and 6; no associated cardiac and extracardiac anomaly was observed. The baby of case 5 had at 3 days of life a supraventricular recurrent tachycardia that required cardiac surveillance but, under arrhythmic drugs, she recovered and presented no other events until 12 months of life.

## 5. Discussion

With an incidence of 0.3–0.5%, PLSVC seems to be a frequent anomaly in the general population; 4–8% of patients with CHD also have PLSVC, as reported in the last two decades. The anomaly is considered benign and without consequences on the cardiac function [7].

However, more recent studies found that 60% of CHD patients have PLSVC, which is also associated with extracardiac anomalies in 37% of cases [8]. The same high rate of cardiac and extracardiac anomalies was reported in Gustapane et al. in a meta-analysis. Moreover, the authors reported a high rate of association between apparently isolated PLSVC and aortic coarctation (21%) [9]. Those findings stress the importance of a third-trimester ultrasound evaluation in cases with isolated PLSVC and, especially, whenever there is a ventricular disproportion or anomalies of the great vessels that draw attention, indicating a possible aortic coarctation. It is well known that PLSVC-associated aortic coarctation morbidity and mortality are higher if not prenatally diagnosed [10].

In our study, PLSVC incidence was 0.31% (six cases in 1897 ultrasound anomaly scans), similar to that reported in the literature. Considering the low association in our group with cardiac and extracardiac anomalies and the absence of aortic coarctation, we suggest that isolated PLSVC is more frequent than thought. The above-mentioned studies reported a higher incidence of association probably due to the postnatal diagnosis of already symptomatic patients, while in asymptomatic patients, isolated PSLVC was overlooked.

Concerning the association of chromosomal anomalies in fetuses with PLSVC, some studies found a rate of 12.5% [8]. Obviously, the correlation is stronger when cardiac and extracardiac anomalies are also present, but Gustapane found that even isolated PLCVS can be associated with aneuploidies in 7% of cases [9].

As a consequence, prenatal invasive diagnosis (at least CMA) is indicated whenever PLCVS is not isolated or whenever the first trimester screening calculated risk was above the cut-off. Chromosomal microarray analysis (CMA) is recommended because it can identify small deletions or duplications that can be missed by a conventional karyotype. A study from 2015 that evaluated the importance of CMA CHD-affected babies found an incremental yield of 7.0% in the overall population of CHD and of 3.4% in isolated cases [11].

In our group, we performed only three genetic analyses in cases that had associated anomalies and were at a higher risk of aneuploidy. No chromosomal anomaly was found, and the follow-up of the babies until up to 2 years did not yield concerns about such associations.

Since 2019, the standardized international and national guideline for second- and third-trimester ultrasound has been of paramount importance in soaring the prenatal detection of CHD. The protocols include two crucial sections also for PLSVC diagnosis: “two-duct” (TD) and “three-vessel and trachea view” (TVT) [12]. These sections show the following landmarks in normal fetuses, from left to right: main pulmonary artery (MPA), aorta and right superior vena cava (RSVC); when a fourth vessel is seen at the left and posterior side of the MPA, there is a clear sign of PLSVC [13]. Additionally, no standardized section can be useful—in longitudinal sections, we can find the “pipe sign” indicated by the drainage of PLSVC in the right atrium through the dilated coronary sinus (CS). A dilated CS identified adjacent to the left atrium wall pathognomonic for diagnosis can be useful and is observed in the four-chamber view (4ChV). CS can routinely be identified and measured in a more posterior section than that obtained with 4ChV, but a CS dilation must be differentiated from a septal defect [14]. We must bear in mind that a dilated CS can lead also to a supracardiac or cardiac total pulmonary vein return anomaly (TPVRA).

Prenatal diagnosis of the PLSVC is not difficult, especially if we use the Doppler effect and analyze the flow direction with a peculiar interest in the innominate vein (IV) [13]. As shown in the cases we presented, the presence of an IV basically excludes a PLSVC diagnosis, though only if the flow direction is correct (from left to right). If the flow is inverted and we have the signs of PLSVC, then RSVC is absent. The image is very tricky to interpret because there are three vessels on the TVT section, but their position is abnormal.

Such a challenging situation was found in case 4, where the fourth vessel was identified on the left side of OPS, but we saw also the IV that theoretically excludes a PLSVC diagnosis. Extending the examination in the sagittal section, we identified a vertical vessel behind the left atria, with a flow directed opposite to the flow in SVC. The vessel collected the pulmonary veins and was directed to the IV. In making the diagnosis, the Doppler effect was crucial in the context of CS normal size and image. The case illustrated how fallacious can be a superficial standard examination of the main sections and the importance of an extended scan whenever an anomaly is found, leading to a correct prenatal diagnosis. This can be life-saving and spare time and invasive late procedures.

In our opinion, for PLSVC prenatal diagnosis, a standard TVT section must be completed with the IV examination in a higher more cranial transverse section. The absence of IV confirms the diagnosis, but its presence does not exclude it in all cases. If the IV has an inverted flow toward the PLSVC, then the RSVC is absent. Other helpful images are to assess the CS where PLSVC drains and also the presence of the RSVC.

Other imaging tools such as computerized tomography, MRI or angio MRI are not used currently in prenatal diagnosis of PLSVC but can be useful in the postnatal period mainly for symptomatic babies and before surgery [15].

Forensic necropsy performed in the same period for intrauterine or peripartum demises did not identify overlooked cases of PLSVC.

Despite the reported association of chromosomal anomalies with PLSVC [16], in the cases where we performed genetic investigation in the same period for different other reasons and the results were abnormal, none of the babies had PLSVC.

Considering the latest studies finding a PLSVC in prenatal anomaly screenings, we must keep searching for intra and extracardiac anomalies, as Minsart et al. showed recently [17]. In a study of 229 fetuses diagnosed with PLSVC, they found only 17% with a true isolated condition; 22% had genetic anomalies, and 41% had both cardiac and extracardiac anomalies. Aortic coarctation was also in that study the most severe and deadly associated condition, but unfortunately seven cases were diagnosed only after birth. As it is shown in the literature, finding PLSVC prompts to extend the investigation of a baby after birth and recommends close surveillance in order to avoid overlooked ominous conditions. In our study group, the last case developed supraventricular tachycardia and required hospitalization, but the evolution was good, and the infant is completely normal at 12 months of life.

The retrospective analysis and the reduced number of cases that did not allow a systematic statistical analysis are the limitations of our study. In order to assess properly the true incidence of PSLVC in normal pregnancies and its association with chromosomal, cardiac and extracardiac anomalies, more studies, on a larger unselected population, are required.

Our study was based on a systematic, extended cardiac examination of fetuses and underlines the importance of not relying only on basic standard sections when an anomaly is found.

## 6. Conclusions

Prenatal diagnosis of PLSVC is feasible and relatively easy if standard protocols are respected at the time of a second-trimester anomaly scan. Once a suspicion of PLSVC is raised, an extended examination of the fetal heart is mandatory. Coronary sinus and innominate vein are clue markers in order to rule out some rare conditions such as the absence of RSVC or TPVRA.

Isolated PLSVC has no significant consequences in utero and in infancy, but its presence can require further cardiac surgery.

## Figures and Tables

**Figure 1 jcm-11-04020-f001:**
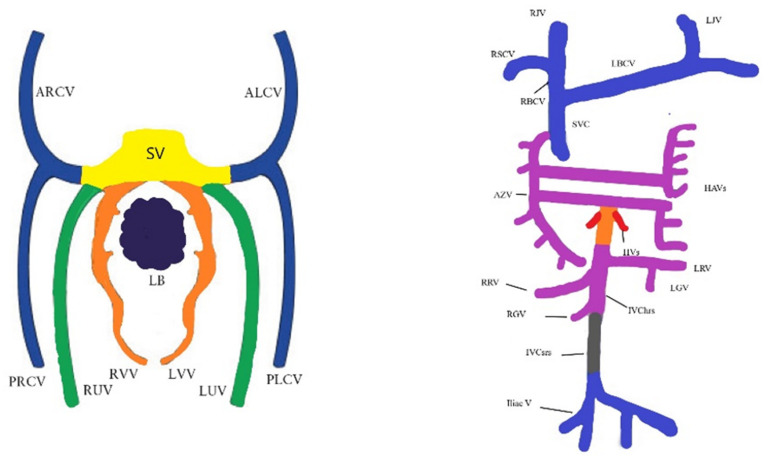
Schematic embryology of the venous system. ALCV, anterior left cardinal vein; ARCV, anterior right cardinal vein; AZV, azygos vein; HAVs, hemiazygos vein; HV(s), hepatic vein(s); Iliac V, iliac veins; IVC hrs, inferior vena cava hepatorenal segment; IVC srs, inferior vena cava sacrorenal segment; LB, liver buds; LBCV, left brachiocephalic vein; LGV, left gonadal vein; LJV, left jugular vein; LRV, left renal vein; LUV, left umbilical vein; LVV, left vitelline vein; PLCV, posterior left cardinal vein; PRCV, posterior right cardinal vein; RBCV, right brachiocephalic vein; RGV, right gonadal vein; RJV, right jugular vein; RRV, right renal vein; RSCV, right subclavian vein; RUV, right umbilical vein; RVV, right vitelline vein; SV, sinus venosus; SVC, superior vena cava. Ref. [3].

**Figure 2 jcm-11-04020-f002:**
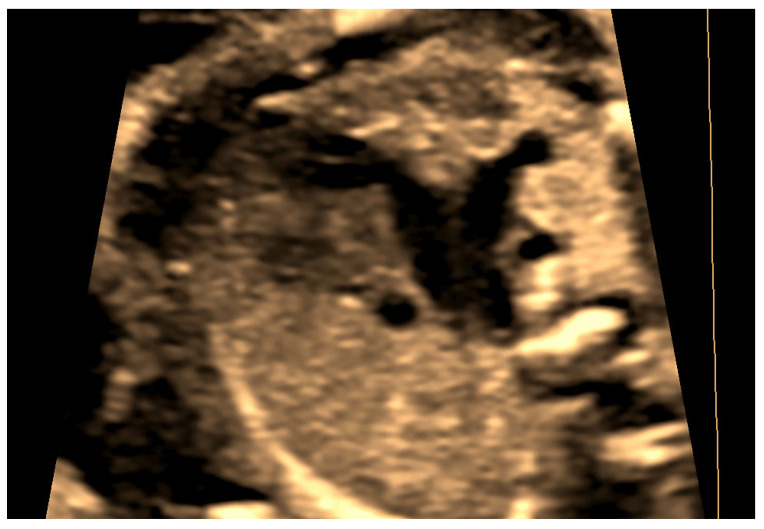
Transversal section at the TTVV—four vessels can be seen—from left to right, persistent left superior vena cava (PLSVC), main pulmonary artery (MPA), aortic arch (AoA) and right superior vena cava (SVC).

**Figure 3 jcm-11-04020-f003:**
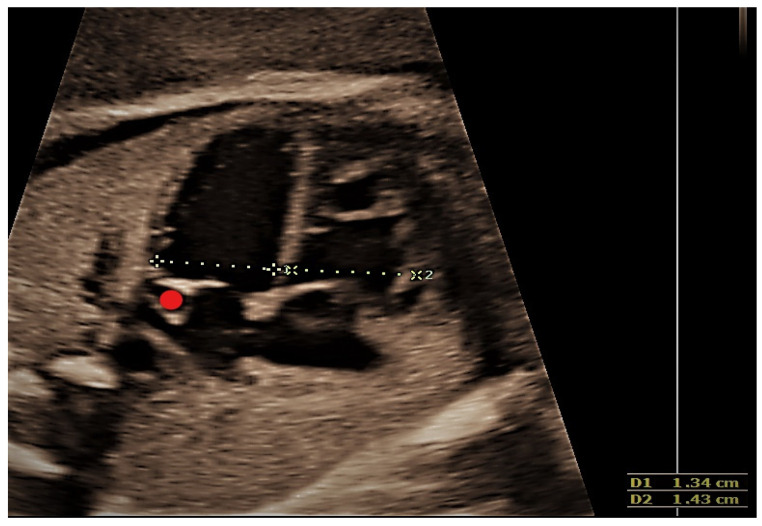
Transversal section of the thorax at 4CV; the red dot marks the dilated coronary sinus in the left atrial wall.

**Figure 4 jcm-11-04020-f004:**
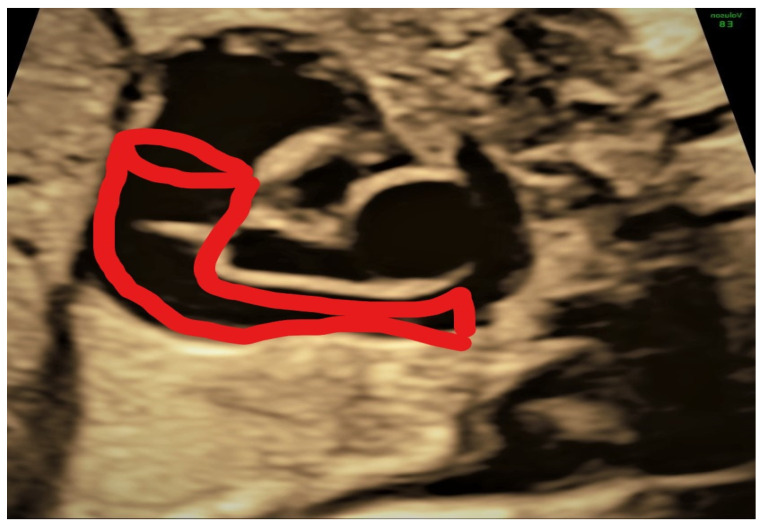
Sagittal section depicting the pipeline aspect of the persistent left superior vena cava (PLSVC).

**Figure 5 jcm-11-04020-f005:**
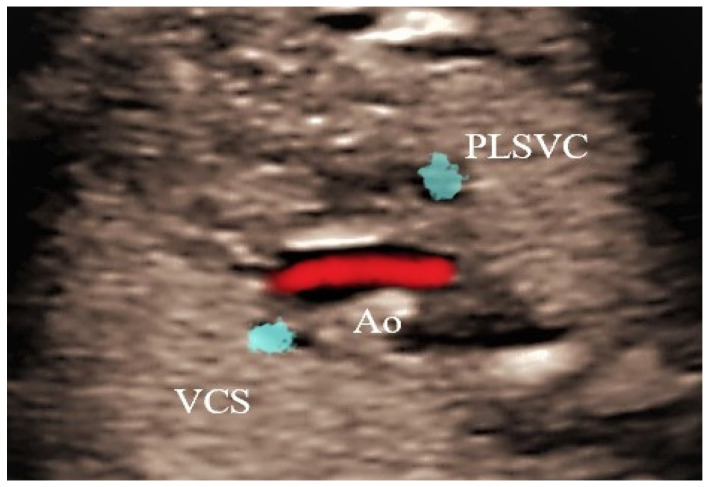
Transversal section superior to the TTVV—absence of the innominate vein. PLSVC—persistent left superior vena cava; Ao—Aorta; VCS—Superior Vena Cava.

**Figure 6 jcm-11-04020-f006:**
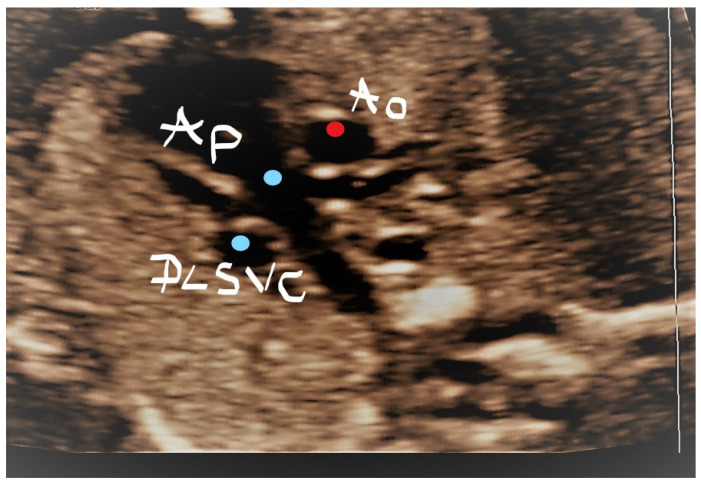
Transversal section at the TTVV level; absence of the right SVC. PLSVC—persistent left superior vena cava; Ao—Aorta; Ap—Pulmonary artery.

**Figure 7 jcm-11-04020-f007:**
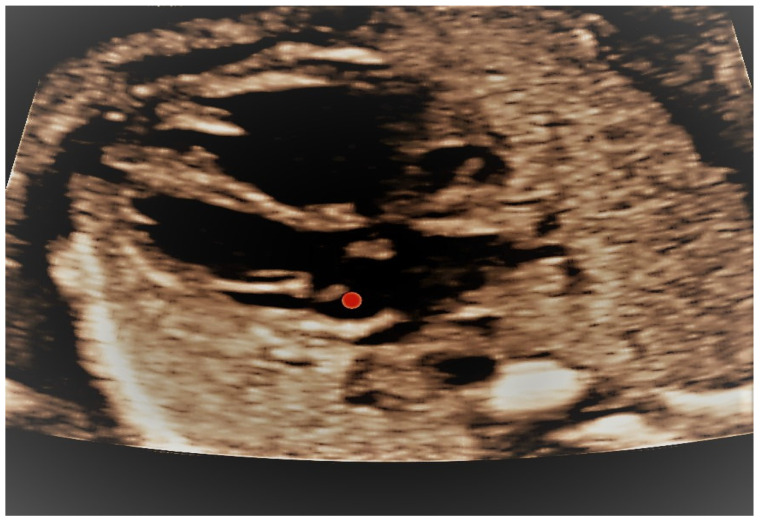
The same case as above, red dot showing a dilated coronary sinus.

**Figure 8 jcm-11-04020-f008:**
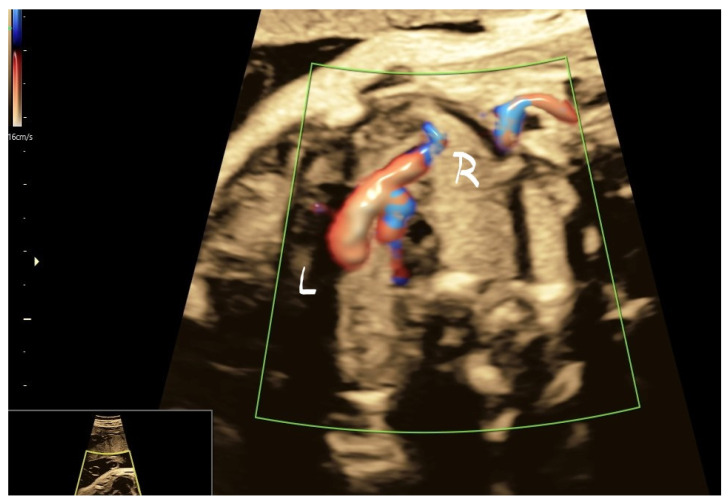
The same case as above; inverted flow in the innominate vein.

**Figure 9 jcm-11-04020-f009:**
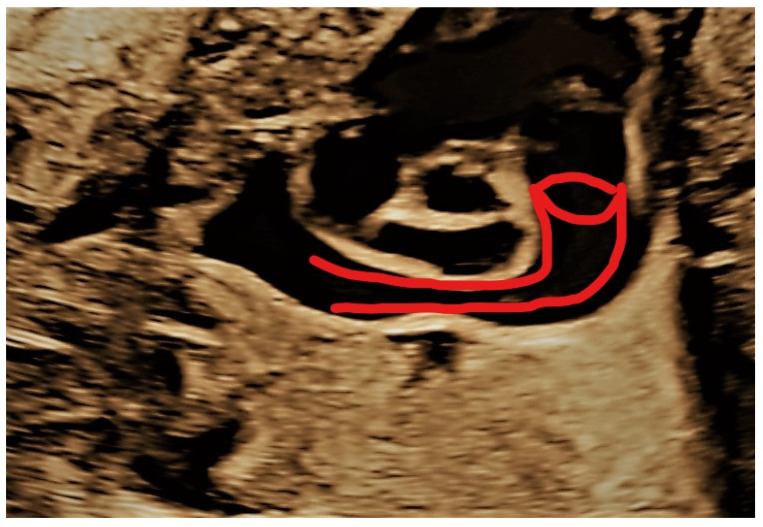
The same case as above; the “pipe sign” sign.

**Figure 10 jcm-11-04020-f010:**
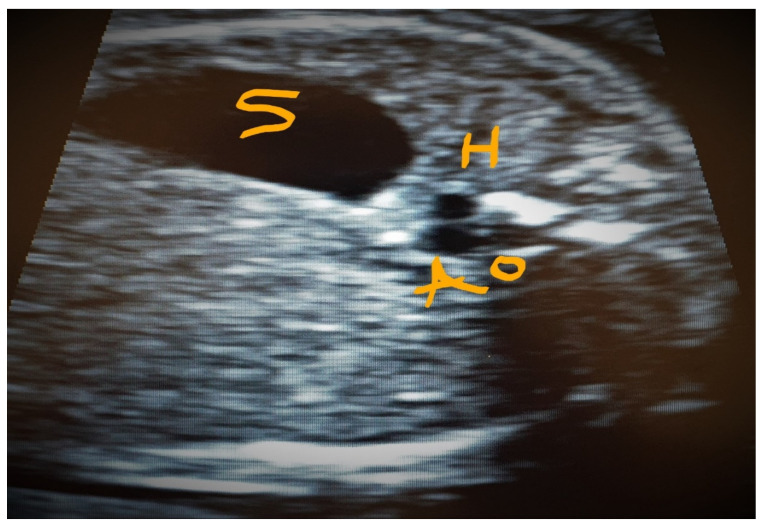
Transverse superior abdominal section showing two vessels side by side, S—Stomach, H—the hemiazygos vein and the aorta Ao.

**Figure 11 jcm-11-04020-f011:**
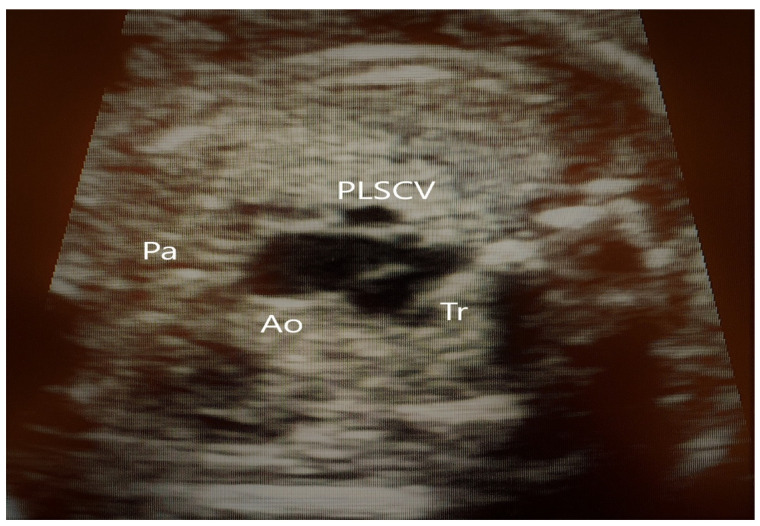
The same case as above; absence of SVC at the TTVV. (PLSVC—persistent left superior vena cava, Ao—Aorta, Pa—Pulmonary artery, Tr—Trachea).

**Figure 12 jcm-11-04020-f012:**
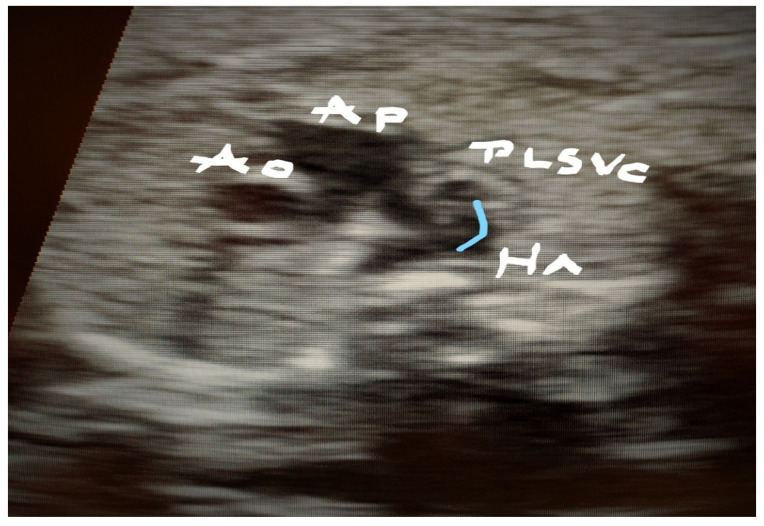
Drainage of the hemiazygos vein in the persistent left superior vena cava (PLSVC). (PLSVC—persistent left superior vena cava, Ao—Aorta, Pa—Pulmonary asrtery, Ha—hemiazygos vein).

**Figure 13 jcm-11-04020-f013:**
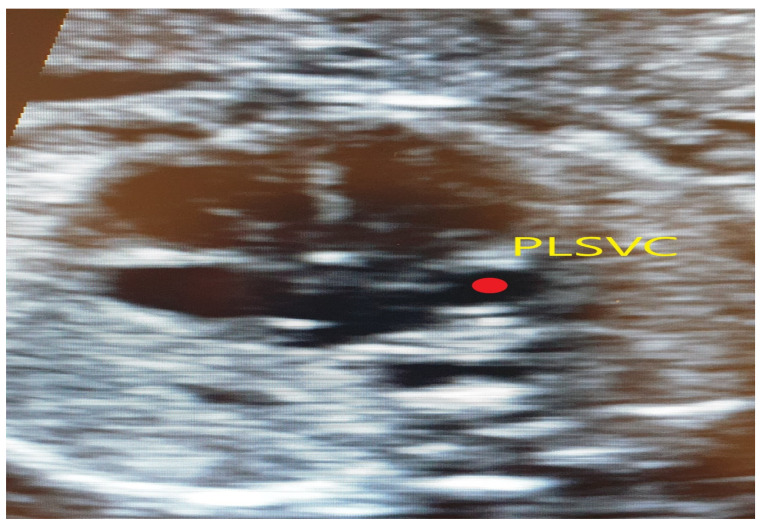
The same case as above; dilated coronary sinus. Persistent left superior vena cava (PLSVC).

**Figure 14 jcm-11-04020-f014:**
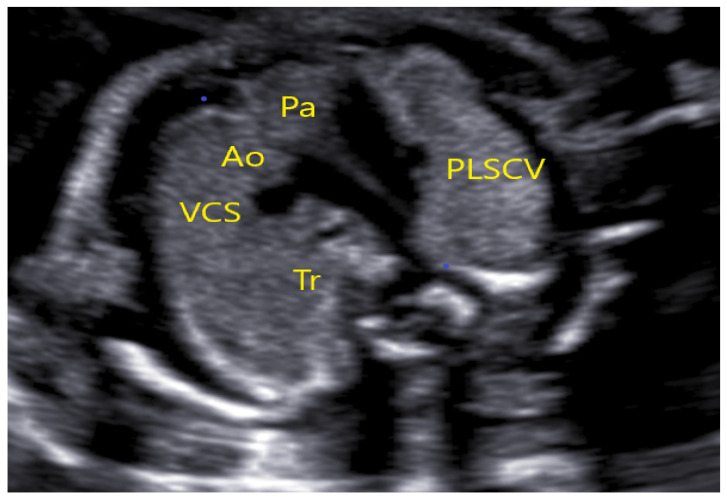
Case 4—TTVV with four vessels. (PLSVC—persistent left superior vena cava, Ao—Aorta, Pa—Pulmonary artery, VCS—Superior Vena Cava, Tr—Trachea).

**Figure 15 jcm-11-04020-f015:**
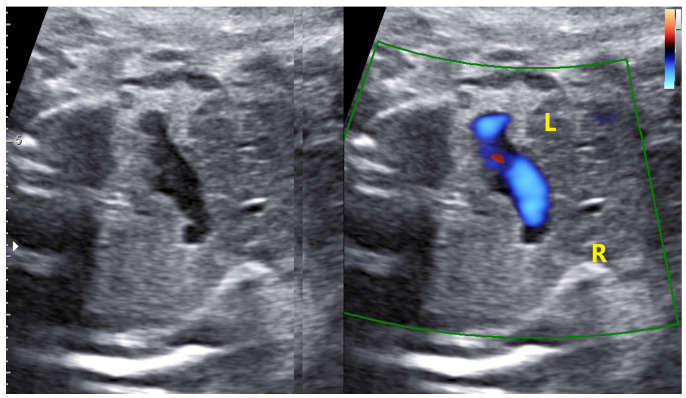
The same case as above; innominate vein with a normal inverted flow.

**Figure 16 jcm-11-04020-f016:**
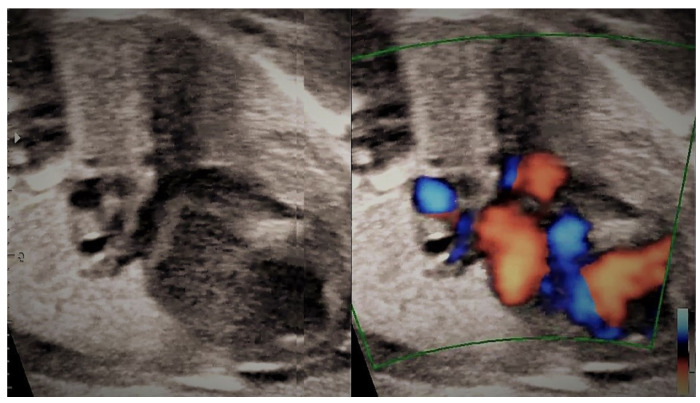
The same case as above; retro atrial vessel behind the left atrium.

**Figure 17 jcm-11-04020-f017:**
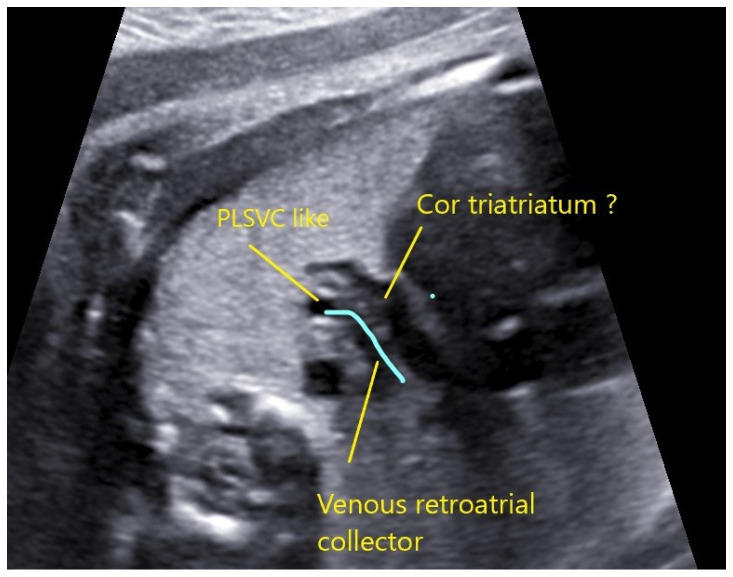
The same case as above; venous ascendant collector behind the left atrium and PLSCV (persistent left superior vena cava).

**Table 1 jcm-11-04020-t001:** Main characteristics of the cases.

Case	Maternal Age	Gestational Age at First Diagnosis	Maternal BMI	Smoking Maternal Status	Significant Hystory	Associated Cardiac Anomalies	Extracardiac Anomalies	Cariotype/CMA	Pregnancy Outcome	Postnatal Outcome
1	23	22 + 1	28.7	No	No	No	Renal dilatation SUA	46XY	Prematurity (36 weeks)	N
2	33	21 + 5	23.2	Yes	No	Dilated CS, absent SVC, inversed flow on IV	No	NP	N	N
3	29	20 + 6	31.4	Yes	Dichorionic twin	Dilated CS, absent SVC, inversed flow on IV, absent infradiaphragmatic IVC and hemiazygos vein	No	NP	Prematurity due to twin pregnancy—35 weeks	Transitory tachypnea not related to venous anomaly
4	34	22 + 1	26.1	No	No	Total pulmonary vein return anomaly	No	46XY	N	Transferred in a cardiac surgery unit for TPVA, at 14 weeks, normal outcome at 9 months
5	31	21 + 2	24.7	No	No	TVPA	No	46XX	N	Supraventricular tachycardia cardiac surveillance, normal at 3 months check, normal at 12 months
6	28	31 + 4	29.1	No	No		No	NP	N	N

## Data Availability

Data are available from the corresponding author.

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
