# Peer review of "Persistent Left Superior Vena Cava Significance in Prenatal Diagnosis—Case Series"

_jcm, 2022, doi:10.3390/jcm11144020_

Round 1
Reviewer 1 Report
1/. All the abbreviations should be explained. What is the difference between PLSVC, LPSVC (65,66) PSLVC (155, 157)
2/. The references. Mostly old. Only seven last decade. 1(281) and 4 (288) references- the same.
3/. Many interpunction mistakes like: 50, 53, 55, 56, 64, 106, 182, 185, 197, 239, 245 and more
4/. Many other mistakes ( missing letters) i.e.123, 226, 251
5/. Please decide which English - British or US - i.e. FOETUS or FETUS
6/. PLSVC IS NOT FREQUENT - is rather rare, but of course, it is probably the most common congenital malformation of thoracic venous return.
I would suggest checking for minor grammatical problems, missing commas, and syntax errors
Author Response
Dear Reviewer,
We really appreciate your effort to evaluate our manuscript. We followed all your recommendations and we performed the changes into the text as follows
1/. All the abbreviations should be explained. What is the difference between PLSVC, LPSVC (65,66) PSLVC (155, 157). We made the corrections.
2/. The references. Mostly old. Only seven last decade. 1(281) and 4 (288) references- the same. We changed
3/. Many interpunction mistakes like: 50, 53, 55, 56, 64, 106, 182, 185, 197, 239, 245 and more. We corrected the errors
4/. Many other mistakes ( missing letters) i.e.123, 226, 251. We corrected the errors
5/. Please decide which English - British or US - i.e. FOETUS or FETUS. We corrected the errors
6/. PLSVC IS NOT FREQUENT - is rather rare, but of course, it is probably the most common congenital malformation of thoracic venous return. We performed the adjustment
I would suggest checking for minor grammatical problems, missing commas, and syntax errors. We checked and changed.
Kind regards,
The authors

Reviewer 2 Report
I read the paper "Persistent Left Superior Vena Cava significance in prenatal diagnosis –cases series". It´s interesting, but I have some suggestions/questions
1.- The introduction is very laconic and authors need to extend. It would be good for the readers to know what environmental or social factors couls be related with PLSVC.
2.- Figure 1 has to be improved
3.- Table 1. It needs more information, about the mother's characteristics. BMI, if they are overweight, obese or underweight, smoking status, education level and even race if possible.
4.- Methods: authors need to better describe the population: where are they from?, from a local place or a big city?, what kind of hospital or clinic is?, race. When the study started and when finished?. It needs a lot of work!
5.- "The aim of this study is to evaluate the incidence and significance of the prenatal diagnosis of PLSVC in our clinic and to review the literature data regarding PLSVC diagnosis and association to cardiac and extracardiac anomalies." I think authors can improve the discussion about the review the literature data regarding PLSVC diagnosis and association to cardiac and extracardiac anomalies. I think authors can include more studies and details.
6.- Is this paper or the results relevant for a general population (other races or countries) or just for the local people where the study was made?
Author Response
Dear Reviewer,
We really appreciate your effort to evaluate our manuscript. We followed all your recommendations, and we performed the changes into the text as follows
1.- The introduction is very laconic and authors need to extend. It would be good for the readers to know what environmental or social factors couls be related with PLSVC.
There are no mentions of social or environmental factors that could be responsible for PLSVC in the literature which otherwise is scarce regarding that condition, mainly as prenatal diagnosis. The postnatal studies as are related to the importance of being aware of that anomaly at the time of cardiac surgery (i.e mitral surgery ) or when it is associate with other cardiac anomalies as heterotaxy, aortic coarctation absence of Right Superior Vena cava and dilated coronary sinus.
2.- Figure 1 has to be improved We adjusted fig 1
3.- Table 1. It needs more information, about the mother's characteristics. BMI, if they are overweight, obese or underweight, smoking status, education level and even race if possible.
We calculated mothers BMI and checked for smoking status in patient’s filed. We did not retained education level since there was no such correlation in the literature. Our population is very homogenous consisting almost exclusively from Caucasian women. The results are inserted in the results and table 1
4.- Methods: authors need to better describe the population: where are they from?, from a local place or a big city?, what kind of hospital or clinic is?, race. When the study started and when finished?. It needs a lot of work!
We made the required changes. As a tertiary maternity centre we receive referred pregnancies from the southern part of Romania (4 administrative departments and usually from Bucharest –meaning that our mainly group of patients are from the urban area. In our country the population is very homogenous consisting mainly from Caucasian people. No other races were encountered in our study. )
5.- "The aim of this study is to evaluate the incidence and significance of the prenatal diagnosis of PLSVC in our clinic and to review the literature data regarding PLSVC diagnosis and association to cardiac and extracardiac anomalies." I think authors can improve the discussion about the review the literature data regarding PLSVC diagnosis and association to cardiac and extracardiac anomalies. I think authors can include more studies and details.
We searched the literature for prenatal diagnosis of PLSVC and found out that the findings of Gustapane in his metanalysis and systematic literature review are accurate, there are not many reports on prenatal diagnosis of PLSVC. In their review Gustapane and all found actually only 13 papers that reffered to prenatal diagnosis of PLSVC gathering 501 cases in 15 years with a high heterogenicity. We added the latest findings in the area published by Minsard and Azzizova.
6.- Is this paper or the results relevant for a general population (other races or countries) or just for the local people where the study was made?
Since the study was conducted in a Romanian tertiary unit the study is relevant for the Romanian general population. This is a limit o the study because we do not have access to a more broad heterogenous population. On the other hand the literature did not mention any difference across the world and related to gender, race or ethnicity.
Kind regards,
The authors

Round 2
Reviewer 1 Report
Dear authrs
I am satisfied with the your responses to my questions raised in initial review.
Reviewer 2 Report
Dear authors,
I agree with the changes. Regards